# Temporal and Spatial Variations in Zebrafish *Hairy/E(spl)* Gene Expression in Response to Mib1-Mediated Notch Signaling During Neurodevelopment

**DOI:** 10.3390/ijms25179174

**Published:** 2024-08-23

**Authors:** Yi-Chieh Chen, Fu-Yu Hsieh, Chia-Wei Chang, Mu-Qun Sun, Yi-Chuan Cheng

**Affiliations:** 1Neuroscience Research Center, Chang Gung Memorial Hospital, Linkou, Taoyuan 333423, Taiwan; yichieh2358@gmail.com; 2Department of Neurology, Chang Gung Memorial Hospital at Linkou Medical Center, Taoyuan 333423, Taiwan; 3Graduate Institute of Clinical Medical Sciences, College of Medicine, Chang Gung University, Taoyuan 33302, Taiwan; 4Graduate Institute of Biomedical Sciences, College of Medicine, Chang Gung University, Taoyuan 33302, Taiwan

**Keywords:** hairy/E(spl), *her* genes, Notch signaling, neural development, zebrafish

## Abstract

Notch signaling is a conserved pathway crucial for nervous system development. Disruptions in this pathway are linked to neurodevelopmental disorders, neurodegenerative diseases, and brain tumors. *Hairy/E(spl) (HES)* genes, major downstream targets of Notch, are commonly used as markers for Notch activation. However, these genes can be activated, inhibited, or function independently of Notch signaling, and their response to Notch disruption varies across tissues and developmental stages. MIB1/Mib1 is an E3 ubiquitin ligase that enables Notch receptor activation by processing ligands like Delta and Serrate. We investigated Notch signaling disruption using the zebrafish Mib1 mutant line, *mib1^ta52b^*, focusing on changes in the expression of *Hairy/E(spl)* (*her*) genes. Our findings reveal significant variability in *her* gene expression across different neural cell types, regions, and developmental stages following Notch disruption. This variability questions the reliability of *Hairy/E(spl)* genes as universal markers for Notch activation, as their response is highly context-dependent. This study highlights the complex and context-specific nature of Notch signaling regulation. It underscores the need for a nuanced approach when using *Hairy/E(spl)* genes as markers for Notch activity. Additionally, it provides new insights into Mib1’s role in Notch signaling, contributing to a better understanding of its involvement in Notch signaling-related disorders.

## 1. Introduction

The vertebrate nervous system encompasses a diverse array of neuronal and glial cell types, which rely on the precise orchestration of cellular processes, including the proliferation and differentiation of progenitor cells. Notch signaling, a highly conserved pathway, is essential for numerous developmental processes, including nervous system development. It plays critical roles in neural stem cell maintenance and neurogenesis in the embryonic brain [1]. Furthermore, evidence suggests a post-developmental role for Notch signaling in the nervous system, with irregularities in the pathway implicated in several neurodegenerative diseases and brain tumors [2,3]. Additionally, aberrant Notch signaling has been associated with various neurodevelopmental disorders, such as autism [4], developmental delay, intellectual disability, and brain malformations [5].

Notch signaling is initiated when Delta, Serrate/Jagged, and Lag-2 transmembrane ligands bind to the Notch receptor. A critical step in this process is the ubiquitination of Notch ligands, facilitated by E3 ubiquitin ligases such as Mind bomb (MIB) and Neuralized (Neur). MIB is essential for the ubiquitination and subsequent endocytosis of ligands like Delta and Serrate, which are necessary for activating the Notch receptor on adjacent cells [6,7]. Loss of MIB function leads to a significant reduction in Notch pathway activation due to the inefficient ubiquitination and internalization of Delta and Serrate/Jagged. As a result, these ligands accumulate on the cell surface, preventing the activation of Notch receptors on neighboring cells and thereby blocking the downstream signaling cascade [7,8]. For readers seeking a visual representation of the Notch signaling pathway, we recommend referring to comprehensive review articles, such as those by Zhou et al. (2022) [9] and Kopan et al. (2009) [10].

Upon ligand–receptor interaction, the Notch receptor undergoes multiple proteolytic cleavages. The first cleavage is mediated by a disintegrin and metalloproteinase (ADAM) family protease at the extracellular domain, followed by a second cleavage within the transmembrane domain by the γ-secretase complex. These cleavages release the Notch intracellular domain (NICD) from the membrane, allowing it to translocate into the nucleus. Once in the nucleus, NICD interacts with Mastermind-like (MAML) proteins, which bridge NICD and the transcription factor CSL [CBF1/RBPjκ, Su(H), Lag-1] [11]. This interaction converts CSL from a transcriptional repressor to an activator, thereby initiating the transcription of Notch target genes, notably the HES [hairy/Enhancer of split, Hairy/E(spl)] family of basic helix–loop–helix (bHLH) transcription factors. The HES proteins, in turn, play a crucial role in repressing the expression of proneural genes, such as *ASCL1*/*Ascl1* and *NEUROG2*/*Neurog2*, which confer neuronal competence and identity. The HES family consists of seven members, each containing a conserved bHLH domain that enables homodimer formation and DNA binding. Additionally, they harbor a WRPW repression domain at their C-terminus, which facilitates the recruitment of co-repressors such as Groucho homologues. Furthermore, they have the capability to form heterodimers with bHLH activators, thereby inhibiting their DNA binding and transcriptional activity [12,13]. Many studies have relied on the upregulation of *HES*/*Hes* gene expression as a dependable indicator of Notch pathway activation. However, despite being primarily regarded as targets of Notch-mediated signals, some *HES*/*Hes* genes have been found to be inhibited or unresponsive to Notch signaling. For instance, HES1/Hes1 is involved in regulatory processes that can occur both in a manner dependent on and independent of Notch signaling [14].

The zebrafish is widely acknowledged as an outstanding vertebrate model for exploring the pathogenesis of human diseases, owing to its transparent embryonic development, ease of breeding, high genetic similarity to humans, and straightforward gene manipulation capabilities [2]. This model has shown promise in replicating the phenotypes of various human genetic disorders. Increasing evidence also highlights the zebrafish as a valuable organism for investigating neurodevelopmental disorders, as it closely resembles humans physiologically and exhibits sensitivity to both pharmacological and genetic interventions [15]. To date, 15 zebrafish hairy/Enhancer of split homologues, named Her (zebrafish homologue of hairy/Enhancer of split related) and Hey (hairy/Enhancer of split related with YRPW motifs), have been identified. They have been implicated in neural development, somitogenesis, and aortic development. Similar to mammals, not all zebrafish *her* genes respond to Notch signaling in the same manner. The regulation of expression for these genes may vary across different developmental stages and among various cell populations, leaving inconclusive answers about their distinct responses to Notch activation. Understanding the dynamics of *HES*/*Hes*/*her* gene expression in response to Notch signaling necessitates techniques such as in situ hybridization, which provide spatial information regarding the regulatory response.

To comprehensively analyze the spatial and temporal expression patterns of *her* genes in response to Notch activation, we isolated all zebrafish *her* genes and investigated their expression in the context of Notch signaling by utilizing the Mib1 mutant embryos (*mib1 ^ta52 b^*), which exhibit a significant perturbation in the Notch pathway due to a mutation in a ubiquitin ligase required for Delta ligand activity [16]. To elucidate their expression dynamics, we conducted in situ hybridization at two critical stages of neural development, allowing us to gather essential spatial information. It is important to note that *mib1 ^ta52 b^* do not represent a complete knockout of all Notch receptors; rather, they model a specific perturbation within the Notch signaling pathway.

## 2. Results

### 2.1. Examination of her Gene Expression in mib1 ^ta52 b^ Mutants

We isolated zebrafish *her* genes, including *her1*, *her2*, *her3*, *her4.1*, *her5*, *her6*, *her7*, *her8 a*, *her8.2*, *her9*, *her11*, *her12*, *her13* (*her13.1*), *hes6* (*hes13.2*), and *her15.1*. These genes were named according to the most recent nomenclature from zfin.org. We categorized these genes into five groups: Hes1, Hes3, Hes5, Hes6, and Hes7, based on their phylogenetic similarities [17,18]. *her1* and *her7* are exclusively expressed in the somites and were not detected in the developing nervous system [19,20,21], and thus not considered in this study. *her2* expression in the *mib1 ^ta52 b^* mutant and wild-type has been previously described [22] and was not reiterated here to avoid redundancy.

The embryos analyzed were generated by in-crossing *mib1 ^ta52 b^* heterozygous parents, and *her* mRNA expression was assessed through in situ hybridization and reverse transcriptase quantitative PCR (RT-qPCR). The genotypes of these embryos were further confirmed using single embryo genotyping after in situ hybridization. To investigate the effect of *mib1 ^ta52 b^* mutation on *her* expression in the developing nervous system, we focused on two developmental stages: the bud stage and 24 h post-fertilization (hpf), marking the stages of neurulation and the initiation and maturation of neurogenesis.

### 2.2. Hes1 Group: her6 and her9

#### 2.2.1. *her6*

In situ hybridization demonstrated that at the bud stage, prominent *her6* transcripts were observed in the anterior neural plate (forebrain primordium) and midline, consistent with previous findings [23]. Additionally, two pairs of lateral proneuronal domains consisting of interneurons and sensory neurons expressing *her6* were observed (Figure 1). No substantial difference in *her6* expression was detected in *mib1 ^ta52 b^* homozygous mutants compared to wild-type siblings, as confirmed using qRT-PCR (Figure 1), indicating that *her6* is not regulated by Mib1-Notch signaling at this developmental stage.

At 24 hpf, wild-type embryos exhibited three stripes of *her6* expression in the posterior forebrain, particularly in a region anterior to the fore-midbrain boundary, with the middle stripe displaying the strongest expression. Surrounding this expression were parallel strips of relatively weaker expression. This region has been identified as the thalamus, where *her6* regulates neuronal identity [24]. Additionally, cells expressing varying levels of *her6* were dispersed throughout the midbrain and hindbrain rhombomeres, with *her6* expression also evident in the hindbrain midline. In *mib1 ^ta52 b^* homozygous mutant embryos, there was a strong induction of *her6* expression in the thalamus, characterized by a robust and broad stripe of expression (Figure 1). Furthermore, both the level of *her6* expression and the number of *her6*-expressing cells were elevated in the midbrain and anterior hindbrain in *mib1 ^ta52 b^* homozygous mutants. However, the arrangement of *her6*-positive cells was notably altered (Figure 1). These results suggest that at 24 hpf, although Notch signaling perturbation leads to an increase in *her6* expression in most *her6*-expressing cells, some *her6*-positive cells lose their *her6* expression upon Mib1-mediated Notch perturbation.

In our study, we observed a strong induction of *her6* expression in the thalamus, while *her9* expression remained unaffected in the *mib1 ^ta52 b^* mutants at 24 hpf. At this stage, the thalamus, part of the diencephalon, is just beginning to emerge and differentiate. In contrast, Sigloch et al. (2023) [25] observed the expression of *her6*, *her9*, *her4*, *her2*, *her8 a*, and *her8.2* in the thalamic complex at 48 hpf. They found that *her6* and *her9* expression was unaffected in *mib1 ^ta52 b^* mutants, whereas the expression of *her2*, *her4.1*, *her12*, *her15*, *hes6*, *her8 a*, and *her8.2* was lost in these mutants. At 48 hpf, the thalamic complex is more structured, and neuronal differentiation is more advanced, with the beginnings of axonal connections. The differing responses of *her6* and *her9* to Notch perturbation between our results and those of Sigloch et al. (2003) in the thalamus at different developmental stages suggest that the regulation of *her* genes by Mib1-mediated Notch signaling is highly dependent on developmental timing.

#### 2.2.2. *her9*

At the bud stage, *her9* exhibited expression in the anterior neural plate, along with longitudinal stripes marking the inter-proneuronal domains between primary motoneurons and interneurons, as previously described (Figure 1) [26,27]. *her9* expression remained unaffected in *mib1 ^ta52 b^* homozygous mutants compared to wild-types, consistent with prior observations (Figure 1) [27].

At 24 hpf, cells expressing varying levels of *her9* were observed in the telencephalon, midbrain, hindbrain rhombomeres and their boundaries, midline of the hindbrain and anterior spinal cord, spinal cord, eyes, and otic placodes (Figure 1). In *mib1 ^ta52 b^* homozygous mutants, *her9* expression was strongly reduced in the mid- and posterior hindbrain and spinal cord, while *her9* expression in the forebrain, midbrain, anterior hindbrain, midline of the hindbrain and spinal cord, and eyes remained unaffected (Figure 1). Conversely, Notch perturbation induced *her9* expression at the mid-hindbrain boundary and otic placodes (Figure 1). *her9* expression in the midline has been shown to promote floor plate development and is independent of Notch signaling [28], as we also found *her9* expression unaffected in *mib1 ^ta52 b^* homozygous mutants. These findings demonstrate that the response of *her9* expression to Notch activation varies depending on the brain regions and cell types.

### 2.3. Hes3 Group: her3

#### *her3* 

Only one zebrafish orthologue, *her3*, is categorized into the Hes3 group based on sequence similarity. The expression pattern of *her3* during early neurogenesis has been documented to occur in the inter-proneuronal domains between primary motoneurons and interneurons (Figure 2). This expression was notably reduced in *mib1 ^ta52 b^* homozygous mutants (Figure 2) [27]. In contrast, a study reported that over-expression of a constitutively active form of Notch (Notch intracellular domain, NICD) represses *her3* expression [29]. These conflicting results may be attributed to the self-inhibitory function of Her3.

At 24 hpf, *her3*-positive neurons were dispersed throughout the midbrain and hindbrain, with relatively higher intensities observed in the anterior midbrain adjacent to the fore-midbrain boundary, anterior hindbrain adjacent to the mid-hindbrain boundary, and spinal cord neurons (Figure 2). This expression was reduced in *mib1 ^ta52 b^* homozygous mutants, with only a few residual *her3*-positive cells remaining in the hindbrain, fore-midbrain, and the mid-hindbrain boundaries (Figure 2). This result suggests that perturbation in Notch signaling is sufficient to downregulate *her3* expression.

### 2.4. Hes5 Group: her4.1, her12 and her15.1

#### 2.4.1. *her4.1*

During early segmentation, *her4.1* is expressed in the proneuronal domains known to be induced by NICD [30], and its expression is reduced in *notch1 a* morpholino knockdown embryos [31]. We observed that at the bud stage, *her4.1* was expressed in the anterior neural plate and longitudinal strips of motor neurons, reticulospinal interneurons, and Rohon–Beard sensory neurons (Figure 3). Additionally, outside of the developing nervous system, *her4.1* was expressed in the presomitic mesoderm. These expressions were repressed in *mib1 ^ta52 b^* homozygous mutants (Figure 3). However, weak residual *her4.1* expression was detected at the most anterior region of the interneuronal stripes (Figure 3).

At 24 hpf, strong and intense expression of *her4.1* could be detected in the forebrain, with cells exhibiting varying degrees of *her4.1* expression in other brain regions, spinal cord, and the eyes (Figure 3). This expression was substantially reduced in *mib1 ^ta52 b^* homozygous mutants; however, a few *her4.1*-positive cells remained clustered in the fore-hindbrain boundary and scattered in the hindbrain (Figure 3). Collectively, these results indicate that Notch activation is essential for *her4.1* expression.

#### 2.4.2. *her12*

At the bud stage, the expression of *her12* could be detected in three longitudinal stripes of proneuronal domains consisting of motor neurons, interneurons, and sensory neurons (Figure 3). *her12* was also expressed in the presomitic mesoderm, where it functions together with *her15* to regulate cyclic gene expression and somitogenesis [32] (Figure 3). The expression in the nervous system was reduced in the *mib1 ^ta52 b^* homozygous mutants, whereas weak expression remnants were observed in the presomitic mesoderm. At 24 hpf, strong *her12* expression was observed in the forebrain, midbrain, hindbrain, spinal cord, and retina, with a gap of no expression in the mid-hindbrain boundary (Figure 3). These expressions were completely abolished in *mib1 ^ta52 b^* homozygous embryos (Figure 3). This result indicates that *her12* expression depends on Notch activation in the developing nervous system.

#### 2.4.3. *her15.1*

At the bud stage, the expression pattern of *her15.1* was similar to that of *her12*, as it could be detected in the brain primordium and the three longitudinal stripes of proneuronal domains (Figure 3), as well as in the presomitic mesoderm [32]. In the *mib1 ^ta52 b^* homozygous mutants, *her15.1* expression was reduced, consistent with previous descriptions [27]; however, a few residual *her15.1*-positive cells were observed in the presomitic mesoderm (Figure 3). At 24 hpf, *her15.1* expression resembled that of *her12*, being present in the forebrain, midbrain, hindbrain, spinal cord, and retina (Figure 3). These expressions were completely lost in *mib1 ^ta52 b^* homozygous embryos, similar to that observed for *her12* expression (Figure 3). This result demonstrates that the expression of *her15.1* relies on Notch activation.

### 2.5. Hes6 Group: her8 a, her8.2, her13 and hes6

#### 2.5.1. *her8 a*

We examined the expression of *her8 a* in response to Mib1-mediated Notch perturbation during early neurogenesis. At the bud stage, *her8 a* was expressed in the primordia of the forebrain, midbrain, and hindbrain. Its expression was slightly downregulated in the *mib1 ^ta52 b^* homozygous embryos, but the areas expressing *her8 a* were not altered (Figure 4), indicating that the reduced expression was due to *her8 a*-positive cells reducing their *her8 a* expression upon Notch perturbation, rather than a reduced number of *her8 a*-positive cells. Previously, we described *her8 a* expression at 24 hpf, where it was expressed in the anterior diencephalon, midbrain, hindbrain, and spinal cord, with no expression in the posterior forebrain [17]. This expression was downregulated in the *mib1 ^ta52 b^* homozygous embryos, with remnants of transcripts at the midbrain and mid-hindbrain boundary (Figure 4). These results demonstrate that *her8 a* expression depends on Notch activation.

#### 2.5.2. *her8.2*

The isolation and expression of *her8.2* have not been previously described. At the bud stage, *her8.2* was expressed in longitudinal strips of motor neurons, reticulospinal interneurons, and Rohon–Beard sensory neurons, as well as in the presomitic mesoderm (Figure 4). Additionally, it was ubiquitously expressed in the entire brain and anterior spinal cord at 24 hpf (Figure 4). This expression at both the bud stage and 24 hpf was strongly inhibited in *mib1 ^ta52 b^* homozygous mutants, where almost no expression could be detected (Figure 4). These findings indicate that the expression of *her8.2* highly depends on Notch activation.

#### 2.5.3. *her13*

At the bud stage, prominent *her13* expression was observed in proneural clusters, consistent with previous findings [33] (Figure 4), and there was no significant difference between the wild-type and *mib1 ^ta52 b^* homozygous mutants (Figure 4). At 24 hpf, strong *her13* expression could be observed in the telencephalon, bilateral longitudinal clusters of neurons flanking the midline of the hindbrain and spinal cord, and transverse stripes in the hindbrain, with relatively weak expression in the rhombomere boundaries and the middle of each rhombomere (Figure 4). Relatively weak expression was also detected in the midbrain and retina. In *mib1 ^ta52 b^* homozygous mutants, the expression was reduced but still remained in the anterior forebrain, midbrain, hindbrain, and retina at low levels (Figure 4), revealing Notch perturbation reduces *her13* expression.

#### 2.5.4. *hes6*

*hes6* has been well characterized in somitogenesis and shown to be regulated by FGF signaling [34,35,36], but its expression in the developing nervous system has not been described. Consistent with previous studies, we demonstrated that *hes6* was expressed in the presomitic mesoderm, which was not altered in the *mib1 ^ta52 b^* homozygous mutants, and no expression in the neural tissues was detected at the bud stage (Figure 4). At 24 hpf, different levels of *hes6* expression were detected throughout the entire nervous system, with strong signals observed in the telencephalon, retina, bilateral longitudinal clusters, transverse stripes of neurons in the hindbrain, and spinal cord (Figure 4). The expression of *hes6* in the retina and mid-hindbrain boundary was robustly upregulated in *mib1 ^ta52 b^* homozygous mutants. In contrast, the expression in the telencephalon, hindbrain, and spinal cord was downregulated (Figure 4). This result demonstrates that *hes6* responds to Notch signaling in different manners depending on the cell types.

### 2.6. Hes7 Group: her5 and her11

#### 2.6.1. *her5*

At the bud stage, *her5* was expressed in the primordium of the midbrain and hindbrain, and this expression was not altered by the *mib1 ^ta52 b^* mutation (Figure 5). A previous study demonstrated that at 24 hpf, *her5* was specifically expressed in the mid-hindbrain boundary, where it interacts with Her11 to repress neurogenesis [37]. This process is independent of Notch signaling, as shown by a chemical inhibitor for Notch intracellular cleavage (DAPT) and *notch1 a* mutants *deadly-seven* [38]. We found that *her5* expression was not affected in *mib1 ^ta52 b^* homozygous mutants (Figure 5), consistent with previous observations, and further indicating that *her5* does not respond to Notch activation or perturbation.

#### 2.6.2. *her11*

*her11* is expressed in the presomitic mesoderm and regulates somitogenesis clock [39]. In the developing nervous system, *her11* expression was very similar to *her5* expression, as they were both expressed in the mid-hindbrain at the bud stage and in the mid-hindbrain boundary at 24 hpf. Notably, there was no significant difference in the expression of *her5* between the wild-type and *mib1 ^ta52 b^* homozygous mutants; however, the expression of *her11* was reduced in *mib1 ^ta52 b^* homozygous mutants (Figure 5). These data suggest that although the expression of *her5* and *her11* is co-localized and regulates the formation of the mid-hindbrain boundary, they respond to Notch signaling in different manners.

## 3. Discussion

In this study, we isolated all zebrafish *her* genes and conducted comprehensive expression analysis to present some uncharacterized expressions of *her* genes, as well as compare them to the published literature. We investigated the impact of Mib1-mediated Notch perturbation on *her* gene expression using the *mib1 ^ta52 b^* mutant and found that the extent of Notch signaling disruption varied across different tissues and developmental stages. MIB1/Mib1 has been shown to ubiquitinate both Delta and Serrate/Jagged ligands, facilitating their internalization and recycling, which is necessary for the efficient activation of Notch signaling. However, the precise locations and developmental stages where MIB1/Mib1 affects Notch signaling have not been systematically described. This underscores the significance of our study, which comprehensively examines the expression of all *her* genes in Mib1 mutants, thereby providing insights into the cells and stages where Mib1 mutation impacts Notch signaling, using these *her* genes as downstream markers.

We discovered that different types and regions of neural cells exhibit varying levels of *her* genes and respond to Mib1-mediated Notch perturbation in different manners. Specifically, *her6*, *her9*, and *hes6* could be positively and negatively responsive to Notch perturbation, highly dependent on the region of the nervous system and their expression patterns. On the other hand, *her4.1*, *her8.2*, *her12*, and *her15.1* expressions were strongly repressed by Notch perturbation, whereas *her3*, *her8 a*, and *her13* displayed reduced expression but with residual remnant expression in certain areas of the nervous system. Additionally, *her5* and *her11* are specifically expressed in the mid-hindbrain boundary, regulating its formation, but only *her11* responds to Notch perturbation. Table 1 provides a detailed summary of *her* gene expression in response to the *mib1 ^ta52 b^* mutation.

Our analysis of zebrafish *her* genes in response to Mib1-mediated Notch perturbation reveals intriguing insights into the complex regulatory mechanisms underlying Notch signaling and its impact on cellular processes. We observed diverse patterns of *her* gene expression changes, reflecting the multifaceted roles of Notch signaling in zebrafish development and homeostasis (Table 1). Notably, certain *her* genes exhibit transcriptional upregulation in response to Notch perturbation, suggesting a compensatory mechanism to counterbalance the loss of Notch signaling. This compensatory upregulation may serve to maintain stemness, inhibit differentiation, or promote alternative cell fate decisions in the absence of Notch-mediated repression. Understanding the specific roles of these upregulated *her* genes in compensating for Notch deficiency warrants further investigation and may provide valuable insights into the mechanisms of cellular adaptation to Notch signaling perturbations. Conversely, we observed transcriptional downregulation of certain *her* genes in response to Notch perturbation. This repression could reflect the loss of Notch-mediated activation or the de-repression of other transcriptional regulators that negatively regulate *her* gene expression in the absence of Notch signaling. The identification of these downregulated *her* genes sheds light on potential downstream effectors of Notch signaling and highlights their importance in mediating Notch-dependent cellular responses. Furthermore, our analysis reveals instances where *her* gene expression remains unchanged despite Notch perturbation. This lack of response may indicate the involvement of compensatory signaling pathways or the presence of alternative regulatory mechanisms that buffer against the effects of Notch loss. Elucidating the factors that maintain *her* gene expression under Notch-perturbed conditions will provide valuable insights into the intricate network of signaling pathways that converge on *her* gene regulation.

In the context of neural development, our study reveals that *her* gene expression patterns are intricately regulated by the Notch signaling pathway, exhibiting both cell-/tissue-specific and stage-specific variations. Rather than simple upregulation or downregulation, our findings indicate that the Mib1-Notch-HES/Her cascade operates in a highly context-dependent manner. This variability highlights the need for further investigation into how the same *Hes*/*her* gene can be differentially regulated by Notch signaling in diverse cellular environments, potentially providing deeper insights into the mechanisms of neural development and the roles of Notch signaling in maintaining the balance between neural progenitor maintenance and differentiation.

With respect to disease pathogenesis, the observed perturbations in *her* gene expression in the *mib1 ^ta52 b^* mutants provide insights into potential mechanisms underlying both early developmental abnormalities and neurodevelopmental disorders associated with Notch signaling dysregulation. Defective neural development due to Notch signaling disruptions can lead to a range of congenital conditions, such as Alagille syndrome, which manifests early in embryonic development [40]. Additionally, dysregulation of Notch signaling has been implicated in neurodevelopmental disorders, including conditions such as autism spectrum disorder [4] and intellectual disability. The mechanisms underlying these disorders are not fully understood, but our data suggest that differential regulation of the same *HES*/*Hes*/*her* genes by Notch signaling may contribute to the diverse phenotypes observed in these conditions. The varied responses of these *her* genes indicate that while some aspects of neurogenesis can be resilient to changes in Notch activity, others are highly susceptible, potentially leading to both structural developmental abnormalities and later-life neurological conditions. This highlights the potential for targeting specific components of the Notch signaling pathway in therapeutic strategies for a wide spectrum of disorders linked to aberrant neurogenesis and neural development.

Most of the previous studies have performed analyses at the bud stage or early neurulation stage and reached conclusions about the response of *her* expression to Notch signaling accordingly. However, our results showed that the expression of *her* genes responds to Notch perturbation in a temporal and spatial manner, which is much more complicated than the previous description. Many studies in mammalian cells, mice, and zebrafish models have utilized *HES*/*Hes*/*her* genes as downstream markers for Notch activation, often relying solely on the information that *HES*/*Hes*/*her* expression responds to Notch activation in a single cell type. Therefore, we suggest caution needs to be taken when using these genes as downstream markers for Notch signaling, considering the heterogeneity in *HES*/*Hes*/*her* gene expression patterns in response to Notch signaling. Factors such as cellular context, developmental stage, and tissue specificity should be taken into consideration.

## 4. Materials and Methods

### 4.1. Ethics Statement

The experimental procedures involving zebrafish were conducted in accordance with the guidelines of the Institutional Animal Care and Use Committee (IACUC) and approved under protocols CGU08–86 and CGU11–118. All efforts were made to minimize suffering and ensure the welfare of the animals.

### 4.2. Fish Maintenance

Zebrafish were maintained according to standard protocols in a recirculating aquatic system with a controlled temperature (28 °C) and a 14 h light/10 h dark cycle. Fish were fed a diet of live *Artemia nauplii* (INVE Aquaculture) and commercial fish food pellets (OTTO, Taiwan).

### 4.3. Mib1 Mutant ta52 b (mib1 ^ta52 b^)

The *mib1 ^ta52 b^* zebrafish line was obtained from Zebrafish International Research Center (Oregon, USA). These mutants exhibit a characteristic phenotype including loss of posterior pigmentation and a curved tail, as described by [16].

### 4.4. Single Embryo Genotyping

Genotyping of individual embryos was performed using established protocols. Briefly, zebrafish embryos were collected at the bud stage or 24 hpf and subjected to DNA extraction using a Gentra Puregene Kit (QIAGEN). Genotyping of the ta52 b allele follows a protocol established by the Zebrafish International Research Center. The ta52 b allele contains a single T-to-G point mutation, leading to the substitution of Methionine with Arginine at amino acid residue 1013. The genotyping process begins with the PCR amplification of the target sequence using the primers 5′-GCACCTGTCAGCTGTGTGGAG-3′ and 5′-GGGCACTTGTATGAAAAATACAGTC-3′. Subsequently, the PCR product is subjected to digestion with the NlaIII restriction enzyme (New England Biolabs). The ta52 b mutation abolishes a recognition site for the NlaIII restriction enzyme, resulting in the creation of restriction fragment length polymorphism. Gel electrophoresis was employed to separate PCR products, and banding patterns were compared to wild-type heterozygous, and homozygous, controls to determine the genotype of individual embryos.

### 4.5. In Situ Hybridization:

In situ hybridization was performed to analyze the spatial expression patterns of target genes in zebrafish embryos. The experimental procedures were meticulously controlled to ensure consistency. Equal numbers of embryos were used in each batch, with controls and *mib1 ^ta52 b^* mutant embryos placed together in the same tube throughout all procedures.

The full coding sequence of each *her* gene was used as a template to create riboprobes through in vitro transcription using RNA polymerase enzymes [T7 (PROMEGA) or SP6 (Roche)] in the presence of digoxigenin-UTP (Roche) to generate antisense riboprobes. These probes were then purified and denatured before hybridization. Embryos were fixed in 4% paraformaldehyde, followed by permeabilization with proteinase K. Prehybridization was performed to reduce non-specific binding, followed by hybridization with the labeled RNA probes overnight at an appropriate temperature. After hybridization, stringent washes were performed to remove unbound probes. Color detection was achieved using nitro blue tetrazolium chloride (Roche) and 5-bromo-4-chloro-3′-indolyphosphate p-toluidine salt (Roche) as substrates for alkaline phosphatase-conjugated antibodies (Roche). The color reaction was monitored under a microscope, allowing visualization of the spatial distribution of the target mRNA expression. The reaction was stopped by washing with Tris-EDTA buffer, and embryos were post-fixed and mounted for imaging. Photos were taken for each embryo and documented until the genotype was confirmed.

The *mib1 ^ta52 b^* mutants were distinguished by characteristic morphological features, including loss of posterior pigmentation and a curved tail, as previously described [16]. Additionally, genotypes of the embryos were confirmed by single embryo genotyping to ensure accuracy. In cases where morphological differences were not apparent, such as embryos at the bud stage, the proportion of altered gene expression was quantified to determine if it adhered to Mendel’s Inheritance Law, with an expected ratio of 3:1 for wild-type versus mutants. Confirmation of genotypes was further performed by single embryo genotyping, thereby ensuring the reliability of the experimental results.

### 4.6. Quantitative Real-Time Reverse-Transcription PCR (qRT-PCR)

In the RT-qPCR procedure, embryos underwent homogenization using the TRIzol reagent (Invitrogen), and total RNA was then isolated using a standard extraction technique. Subsequently, cDNA synthesis from the total RNA was achieved utilizing random hexamer priming and the RevertAid First Strand cDNA Synthesis Kit (Bionovas). qPCR analysis was carried out employing the ABI StepOne Real-Time PCR System (Applied Biosystems) along with the SYBR green fluorescent dye (Bionovas). To ensure accurate assessment, gene expression levels were normalized to *gapdh* expression levels and evaluated utilizing the comparative CT method (40 cycles) in accordance with the manufacturer’s instructions (Applied Biosystems).

For statistical evaluation, Student’s *t* test was utilized within Microsoft Excel 2016, with the significance level defined as *p* < 0.05. Each experiment was independently repeated at least three times.

## Figures and Tables

**Figure 1 ijms-25-09174-f001:**
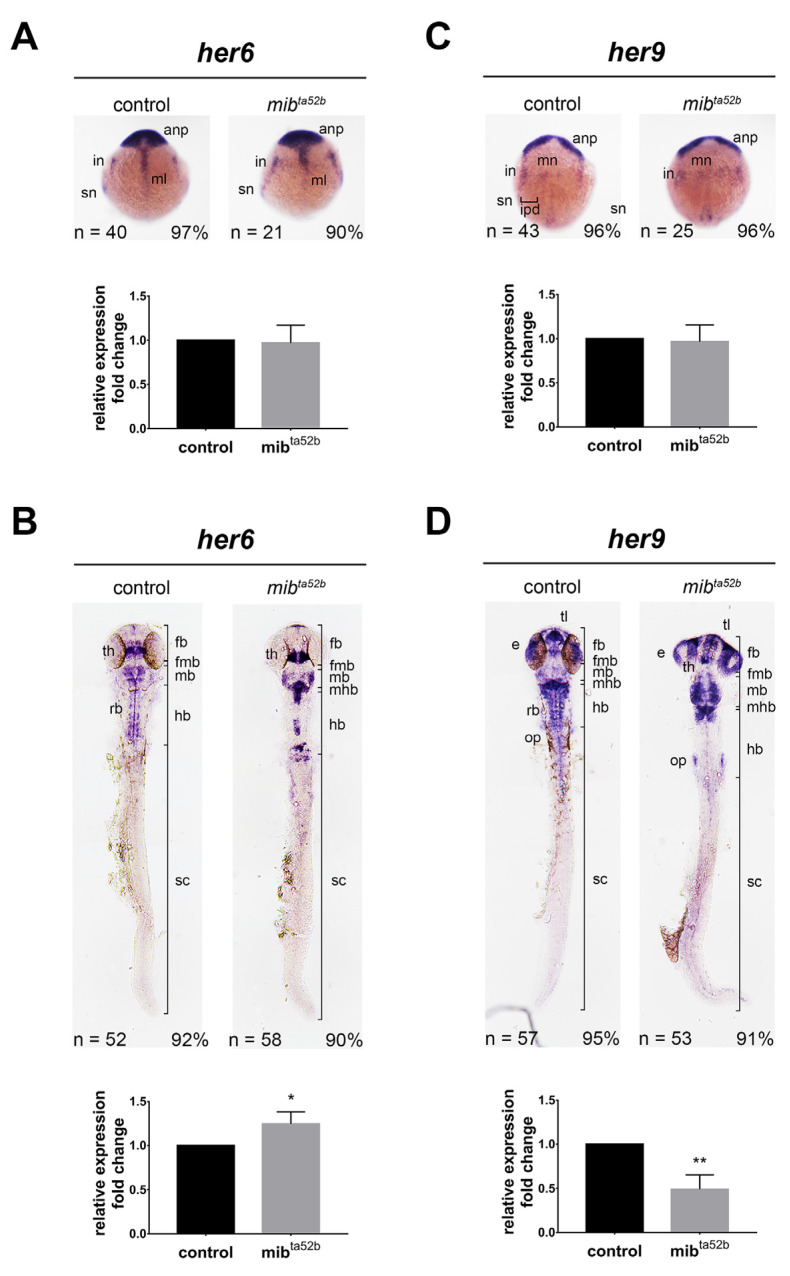
Comparison of *her6* and *her9* expression between wild-type sibling and *mib1 ^ta52 b^* mutant zebrafish.In situ hybridization was conducted to assess the spatial expression of *her6* and *her9* mRNA in wild-type and *mib1 ^ta52 b^* homozygous embryos. (**A**) *her6* expression at the bud stage, (**B**) *her6* expression at 24 h post-fertilization (hpf), (**C**) *her9* expression at the bud stage, and (**D**) *her9* expression at 24 hpf. Embryos are depicted in dorsal views with anterior to the top. To facilitate imaging, the yolk of 24 hpf embryos was removed and flat-mounted. Key anatomical features include the anp, anterior neural plate; e, eye; fb, forebrain; fmb, fore-midbrain boundary; hb, hindbrain; in, interneurons; ipd, inter-proneuronal domain; mb, midbrain; mhb, mid-hindbrain boundary; ml, midline; mn, motoneurons; op. otic placode; rb, rhombomeres; th, thalamus; sn, sensory neurons. The percentages in each panel indicate the proportion of embryos displaying the same phenotype as that shown in the photographs of the total embryos examined. The column charts at the bottom of each in situ hybridization photo represent the quantification of mRNA expression using qRT-PCR. Quantitative data are presented as mean ± standard error of mean (SEM); statistical analysis was performed by Student’s *t*-test. All reactions were performed in triplicate for each sample. *, *p* < 0.05; **, *p* < 0.01.

**Figure 2 ijms-25-09174-f002:**
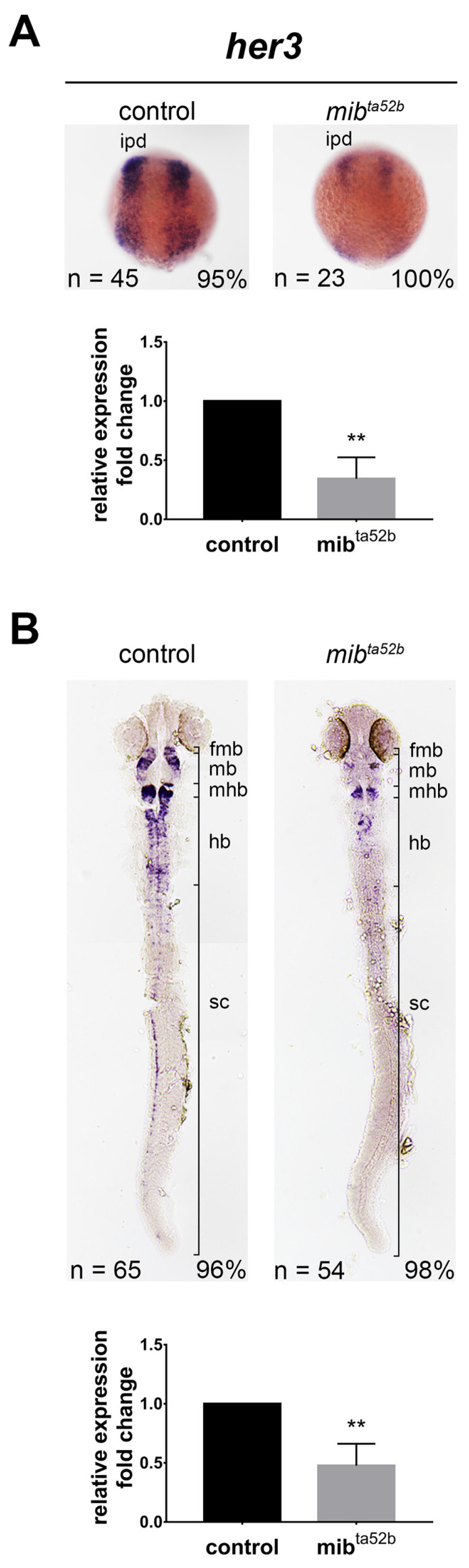
Expression of *her3* in wild-type and *mib1 ^ta52 b^* mutant zebrafish embryos. In situ hybridization demonstrates the expression of *her3* in wild-type siblings (left panels) and *mib1 ^ta52 b^* mutant embryos (right panels) at the bud stage (**A**) and 24 h post-fertilization (hpf) (**B**). Images depict dorsal views with anterior to the top. fmb, fore-midbrain boundary; hb, hindbrain; ipd, inter-proneuronal domain; mb, midbrain; mhb, mid-hindbrain boundary; scn, spinal cord neurons. The expression levels of *her3* were validated using quantitative real-time polymerase chain reaction (qRT-PCR), displayed at the bottom panels. Quantitative data are presented as mean ± standard error of mean (SEM). Statistical analysis was performed by Student’s *t*-test. All reactions were performed in triplicate for each sample. **, *p* < 0.01.

**Figure 3 ijms-25-09174-f003:**
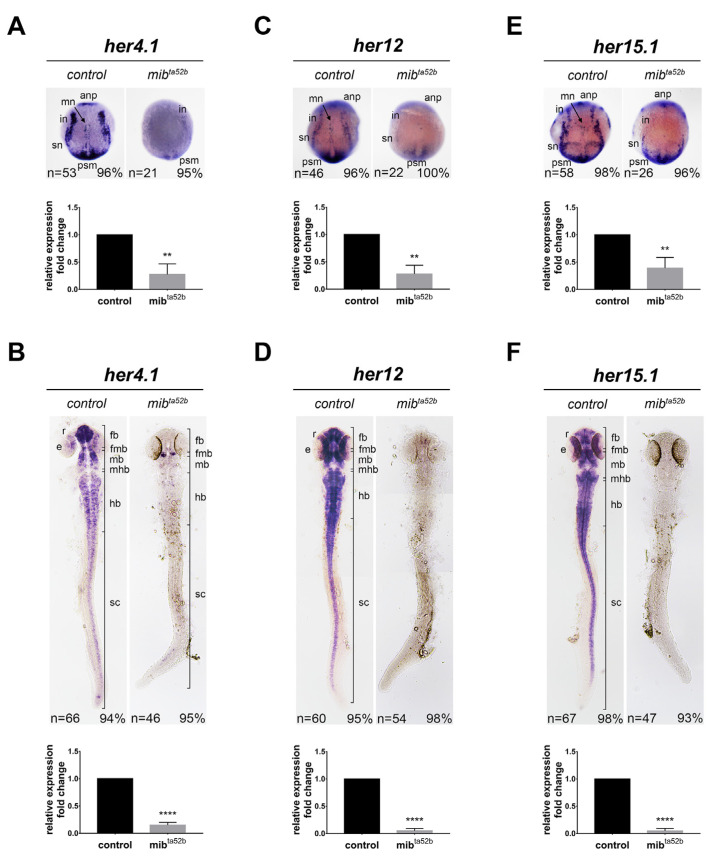
Comparison of *her4.1*, *her12*, and *her15.1* expression in wild-type and *mib1 ^ta52 b^* mutant embryos. The expression of *her4.1* (**A**,**B**), *her12* (**C**,**D**), and *her15.1* (**E**,**F**) was detected in wild-type sibling embryos (left panels) and *mib1 ^ta52 b^* homozygous mutant zebrafish embryos (right panels) at the bud stage (**A**,**C**,**E**) and 24 hpf (**B**,**D**,**F**) using in situ hybridization. Dorsal views with anterior toward the top. Key anatomical features include the anp, anterior neural plate; e, eye; fb, forebrain; fmb, fore-midbrain boundary; hb, hindbrain; in, interneurons; mn, motor neurons; psm, presomitic mesoderm; r, retina; sn, sensory neurons; sc, spinal cord. The expression levels were further validated using quantitative real-time polymerase chain reaction (qRT-PCR). Quantitative data are presented as mean ± standard error of mean (SEM); statistical analysis was performed by Student’s *t*-test. All reactions were performed in triplicate for each sample. **, *p* < 0.01, ****, *p* < 0.0001.

**Figure 4 ijms-25-09174-f004:**
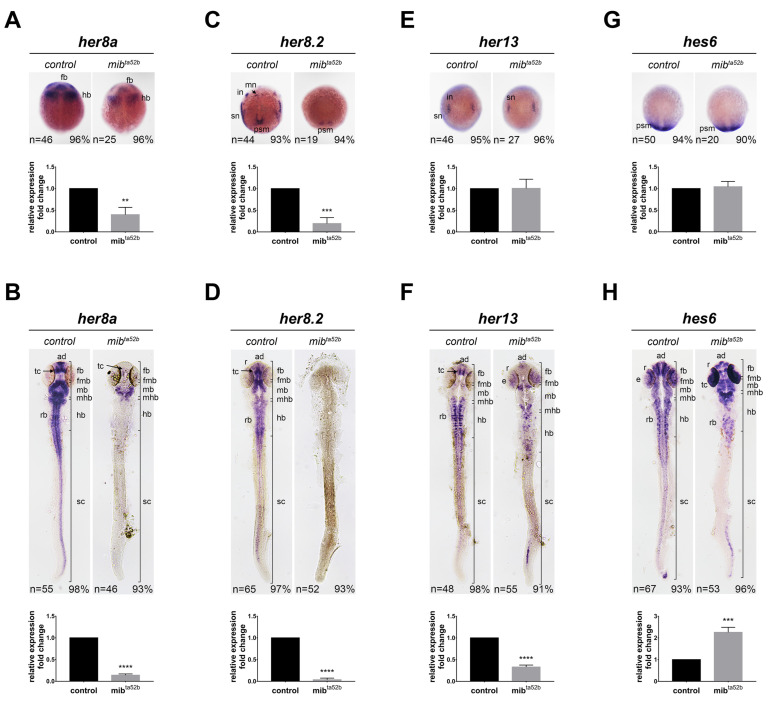
Comparison of *her8 a*, *her8.2*, *her13*, and *hes6* mRNA expression in wild-type and *mib1 ^ta52 b^* mutant embryos by in situ hybridization. The expression of *her8 a* (**A**,**B**), *her8.2* (**C**,**D**), *her13* (**E**,**F**), and *hes6* (**G**,**H**) were detected in wild-type siblings (left panels) and *mib1 ^ta52 b^* homozygous mutant embryos (right panels) at the bud stage (**A**,**C**,**E**,**G**) and 24 hpf (**B**,**D**,**F**,**H**). Dorsal views with anterior toward the top. ad, anterior diencephalon; fb, forebrain; hb, hindbrain; in, interneurons; mb, midbrain; mhb, mid-hindbrain boundary; mn, motor neurons; psm, presomitic mesoderm, r, retina; rb, rhombomeres; rob, rhombomere boundaries, tc, telencephalon; sn, sensory neurons; sc, spinal cord. The expression levels were validated using quantitative real-time polymerase chain reaction (qRT-PCR). Quantitative data are presented as mean ± standard error of mean (SEM). Statistical analysis was performed by Student’s *t*-test. All reactions were performed in triplicate for each sample. **, *p* < 0.01; ***, *p* < 0.001; ****, *p* < 0.0001.

**Figure 5 ijms-25-09174-f005:**
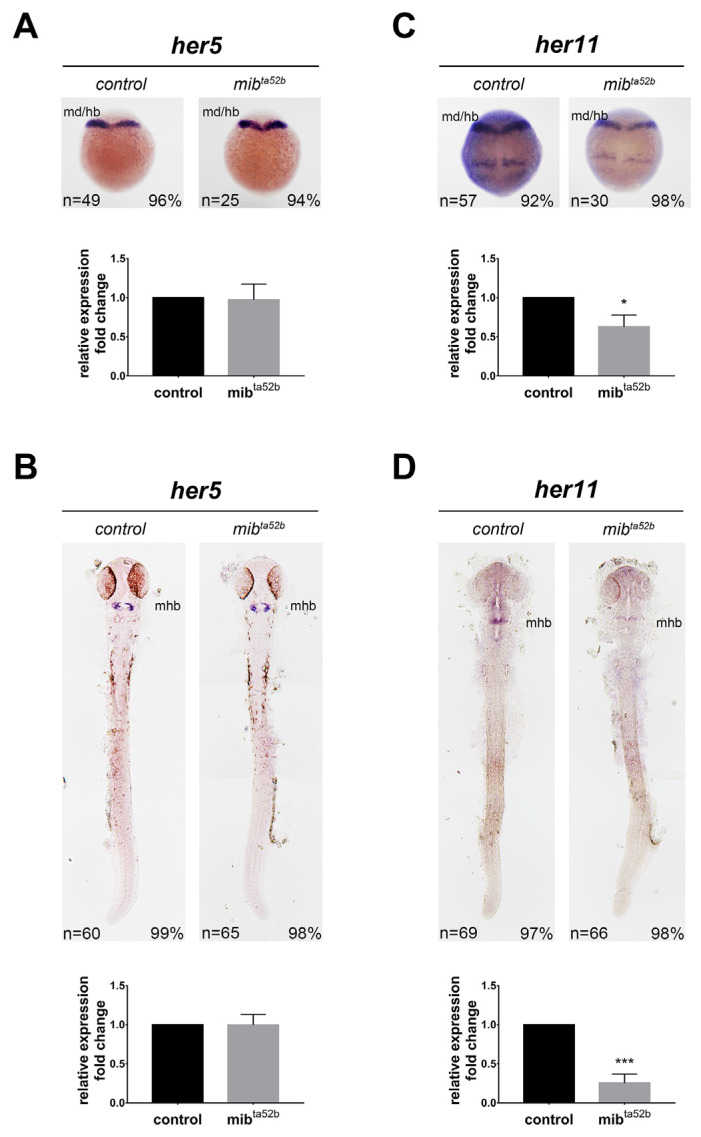
Expression of *her5* and *her11* in wild-type and *mib1 ^ta52 b^* mutant embryos at the bud and 24 hpf stages. The expression of *her5* (**A**,**B**) and *her11* (**C**,**D**) was detected in wild-type siblings (left panels) and *mib1 ^ta52 b^* homozygous mutants (right panels) at the bud (**A**,**C**) and 24 hpf (**B**,**D**) stages. Dorsal views with anterior toward the top. hb, hindbrain; mb, midbrain; mhb, mid-hindbrain boundary. qRT-PCR results show the quantification of expression levels. Quantitative data are presented as mean ± standard error of mean (SEM). Statistical analysis was performed by Student’s *t*-test. All reactions were performed in triplicate for each sample. *, *p* < 0.05; ***, *p* < 0.001.

**Table 1 ijms-25-09174-t001:** Response of her genes to *mib1^ta52 b^* mutation.

*her* Gene	Developmental Stage	Expression in Wild-Type CNS	Expression in *mib1^ta52b^* Mutants	Dependence on Mib1-Mediated Notch Activation
*her6*	Bud Stage	forebrain primordium, midline, interneurons and sensory neurons	Unaltered	Stage-specific response
	24 hpf	thalamus, midbrain, hindbrain	Strong induction	
*her9*	Bud Stage	anterior neural plate, inter-proneuronal domains	Unaltered	Stage-specific response and region-specific response
	24 hpf	telencephalon, thalamus, midbrain, hindbrain, spinal cord, eyes, otic placodes	Expression varies as up, down, or unchanged across regions	
*her3*	Bud Stage	inter-proneuronal domains	Reduced, with residual expression	Yes
	24 hpf	midbrain and hindbrain, mid-hindbrain boundary, spinal cord	Reduced, with residual expression	
*her4.1*	Bud Stage	anterior neural plate, proneuronal domains	Reduced, with residual expression	Yes
	24 hpf	forebrain, midbrain, hindbrain, spinal cord, eyes	Strongly repressed	
*her12*	Bud Stage	anterior neural plate, proneuronal domains, presomitic mesoderm	Reduced, with residual expression	Yes
	24 hpf	forebrain, midbrain, hindbrain, spinal cord, retina	Strongly repressed	
*her15.1*	Bud Stage	anterior neural plate, proneuronal domains	Reduced	Yes
	24 hpf	forebrain, midbrain, hindbrain, spinal cord, retina	Strongly repressed	
*her8a*	Bud Stage	forebrain, midbrain, hindbrain	Reduced, with residual expression	Yes
	24 hpf	anterior diencephalon, midbrain, hindbrain	Reduced, with residual expression	
*her8.2*	Bud Stage	proneuronal domains, presomitic mesoderm	Strongly repressed	Yes
	24 hpf	entire brain, anterior spinal cord	Strongly repressed	
*her13*	Bud Stage	proneural clusters	Unaltered	Stage-specific response
	24 hpf	forebrain, hindbrain	Reduced, with residual expression	
*hes6*	Bud Stage	presomitic mesoderm	Unaltered	Stage-specific response and region-specific response
	24 hpf	entire central nervous system	Expression varies as up, down, or unchanged across regions	
*her5*	Bud Stage	primordium of midbrain and hindbrain	Unaltered	No
	24 hpf	mid-hindbrain boundary	Unaltered	
*her11*	Bud Stage	primordium of midbrain and hindbrain, presomitic mesoderm	Reduced	Yes
	24 hpf	mid-hindbrain boundary	Reduced	

## Data Availability

The data that support the findings of this study are available upon request from the authors.

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
