# Peer review of "Temporal and Spatial Variations in Zebrafish Hairy/E(spl) Gene Expression in Response to Mib1-Mediated Notch Signaling During Neurodevelopment"

_ijms, 2024, doi:10.3390/ijms25179174_

Round 1

Reviewer 1 Report

Comments and Suggestions for Authors

This study addresses the expression of zebrafish members of the HES family upon Notch signalling perturbation, using mib mutants. The response of the Notch target genes of the HES genes to Notch signalling perturbation has been extensively studied for decades. However, the response in each tissue, in different Notch pathway mutants, and in different animal model species, is still not completely resolved. As such, the study may be of value. However, there are several reasons why I am hesitant regarding the current study.

Major issues:

11)      The paper is overinterpreting the findings, and making constant referral to studying “Notch-deficient” mutants. mib is not a “Notch-deficient” line, mib is one of many Notch signal transduction pathways and also not the only ubiquitin ligase involved in Notch ligand function. Hence, mib is a “Notch signalling perturbed” line and the paper presents the expression of Her genes in a mib mutant, not in a “Notch-deficient” mutant. More than likely, the Her expression results would be quite different in a complete Notch signalling null background i.e., a deletion of all Notch receptors.

22)      Relatedly, the Sigloch et al. Development, 2023 paper (PMID 37009986), also addresses Her gene expression in zebrafish mib mutants, and contains many other related Notch-Her experiments. Yet, Sigloch et al. is not referenced or discussed.

Minor issues:

33)      Introduction is somewhat oversimplified and should mention the multiple proteolytic cleavages of the Notch receptor, the processing of the ligands, in particular the ubiquitination of Delta/Ser/Jagged by the mib/neur ligases, especially since they are using a mib mutant in the paper. Also, they fail to mention one key transcriptional co-factor for NICD, namely mastermind.

44)      Introduction and Results: the extent of Notch signalling disruption in mib mutants should be discussed. Where, when and to what extent does mib affect Notch signalling? Mutants for mib is clearly not the same as knocking out all Notch receptors.

55)      Figure legends should state that the graphs are showing the qRT-PCR results.

Comments on the Quality of English Language

OK.

Author Response

Please also see the attachment.

Reviewer #1

“This study addresses the expression of zebrafish members of the HES family upon Notch signalling perturbation, using mib mutants. The response of the Notch target genes of the HES genes to Notch signalling perturbation has been extensively studied for decades. However, the response in each tissue, in different Notch pathway mutants, and in different animal model species, is still not completely resolved. As such, the study may be of value. However, there are several reasons why I am hesitant regarding the current study.”

Major issues:

Comment 1

"11) The paper is overinterpreting the findings, and making constant referral to studying “Notch-deficient” mutants. mib is not a “Notch-deficient” line, mib is one of many Notch signal transduction pathways and also not the only ubiquitin ligase involved in Notch ligand function. Hence, mib is a “Notch signalling perturbed” line and the paper presents the expression of Her genes in a mib mutant, not in a “Notch-deficient” mutant. More than likely, the Her expression results would be quite different in a complete Notch signalling null background i.e., a deletion of all Notch receptors."

Response:

Thank you for highlighting this important distinction. We agree that describing the Mib mutant line as “Notch-deficient” overstates the scope of the perturbation and is not accurate. We have revised the manuscript to refer to the Mib line as a “Notch signaling perturbed” model (correcting 30 related instances), including changes to the title and short running title. Additionally, we have emphasized this point in the introduction (page 2, lines 128-130).

Regarding the differences in her gene expression between Mib mutants and a complete knockout of all four Notch receptors, we have thoroughly addressed this point in our response to Comment 4 (please see the detailed explanation below). In that section, we discussed the potential differences in her gene expression that might arise in a complete Notch signaling null background.

To provide a clearer summary of our findings and address the distinction between Mib mutants and Notch-deficient models, we have added a table to the manuscript (Table 1). This table details "Expression in mib1ta52b Mutants" and "Response to Mib1-mediated Notch Activation," clarifying the specific impact of the Mib1 mutation and distinguishing it from a complete Notch deficiency.

Comment 2

"22) Relatedly, the Sigloch et al. Development, 2023 paper (PMID 37009986), also addresses Her gene expression in zebrafish mib mutants, and contains many other related Notch-Her experiments. Yet, Sigloch et al. is not referenced or discussed."

Response:

We apologize for this oversight. We have now included a discussion of the findings from Sigloch et al. (2023) in our revised manuscript and compared our results with theirs to provide a more comprehensive understanding of her gene expression in Mib1 mutants (Page 6, lines 185-196).

Minor issues:

Comment 3

"33) Introduction is somewhat oversimplified and should mention the multiple proteolytic cleavages of the Notch receptor, the processing of the ligands, in particular the ubiquitination of Delta/Ser/Jagged by the mib/neur ligases, especially since they are using a mib mutant in the paper. Also, they fail to mention one key transcriptional co-factor for NICD, namely mastermind."

Response:

Thank you for this suggestion. We have expanded the introduction by adding two paragraphs to provide a more detailed description of the Notch signaling pathway. This includes an explanation of the multiple proteolytic cleavages of the Notch receptor, the role of Mib/Neur ligases in ligand ubiquitination, and the essential function of mastermind as a transcriptional co-factor for NICD (page 3, lines 66-77 and 79-87).

Comment 4

"44) Introduction and Results: the extent of Notch signalling disruption in mib mutants should be discussed. Where, when and to what extent does mib affect Notch signalling? Mutants for mib is clearly not the same as knocking out all Notch receptors."

Response:

To address the reviewer’s question regarding where, when, and to what extent MIB1/Mib1 affects Notch signaling, and the differences between MIB1/Mib1 mutants and a complete knockout of all four Notch receptors, we conducted an extensive literature search on PubMed using the keywords "Mind bomb + Notch" and "MIB1 + Notch." We retrieved 80 and 115 references, respectively. After thoroughly reviewing these articles, we found no descriptions directly comparing MIB1/Mib1 mutants to a knockout of all four Notch receptors. From these studies, we concluded that since MIB1/Mib1 is responsible for the ubiquitination of multiple ligands (e.g., Delta and Serrate), its deficiency can impair the activation of all four Notch receptors to varying degrees. Additionally, because MIB1/Mib1 deficiency acts upstream of receptor activation, it serves as a broader and more generalized disruptor of the Notch pathway compared to receptor-specific deficiencies. This upstream role explains why MIB1/Mib1 deficiency can produce phenotypes that may overlap with, but are not identical to, those caused by the loss of individual or multiple Notch receptors.

To explore this further, we considered the affinity of Delta and Serrate/Jagged ligands for different Notch receptors, given that MIB1/Mib1 regulates the internalization of these ligands. Notably, the interaction between Delta and Serrate/Jagged ligands and Notch receptors is complex. Delta1, Delta3, and Delta4 are the primary Delta ligands in vertebrates, and their interactions with Notch receptors are not uniform, as different Notch receptors exhibit varying affinities for these ligands. For instance, Notch1 has a high affinity for Delta1, Notch2 shows a preference for Delta4, Notch3 predominantly interacts with Delta4, and Notch4 interacts with both Delta1 and Delta4. Similarly, Notch1 has a high affinity for Jagged1, Notch2 strongly interacts with Jagged1, Notch3 can interact with both Jagged1 and Jagged2, and Notch4 has a lower affinity for Jagged ligands compared to Notch1 and Notch2. However, these regulatory mechanisms are not restricted to the developing nervous system and may not directly apply to our study’s findings. Therefore, the question of where, when, and to what extent MIB1 affects Notch signaling cannot be fully addressed solely by examining the ubiquitination targets of MIB1, namely Delta and Serrate/Jagged.

In conclusion, we generally summarize that the extent to which MIB1/Mib1 affects Notch signaling is complex and currently inconclusive. As the reviewer suggested, we have adopted the term "MIB1 mutation perturbed Notch signaling" throughout the entire article. The uncertainty surrounding the role of MIB1 in Notch signaling further underscores the value of this study, as it provides additional insights that partially address this question. We have included this discussion on page 14, lines 514-524.

Comment 5

"55) Figure legends should state that the graphs are showing the qRT-PCR results."

Response:

We have confirmed that all figure legends clearly state that the graphs are showing qRT-PCR results.

Reviewer 2 Report

Comments and Suggestions for Authors

Abstract: it was a good idea to summarize each sections of the paper in the abstract and gives a good overview about the project, however, the importance of understanding this signalling pathway is missing and it does not raise the interest of the potential reader to this paper. Thus, I suggest a significant re-writing and add some catchy pieces of information.

Introduction: a good and detailed general introduction was given about the Notch signalling cascade. Since the described phenomenon is complex, I recommend to add a schematic figure about it to help your reader.

Materials and methods: a very detail-oriented part of the paper. No big issues were detected, however, please add the manufacturer of each chemical used (as it was nicely done in Section 4.6).

Results: a good description was given. Regarding figures, please use Figure 1a, 1b, 1c, etc to help your reader. Figure legends needed to be fixed, since in the current form it is hard to discriminate if it belongs to the text, or to the figure. As some genes show different expression pattern between brain parts, please do site-specific gene expression analysis.

Discussion: this is the weak point of your manuscript. Now a general overview is given about the results, but we do not gain any piece of information how these findings are integrated into the current knowledge about the topic. The importance of these findings are also missing (i.e: their role in brain development or in disease pathogenesis).

Author Response

Please also see the attachment.

Reviewer #2

Comment 1

"Abstract: it was a good idea to summarize each sections of the paper in the abstract and gives a good overview about the project, however, the importance of understanding this signalling pathway is missing and it does not raise the interest of the potential reader to this paper. Thus, I suggest a significant re-writing and add some catchy pieces of information."

Response:

Thank you for your feedback. We have comprehensively revised the abstract to better emphasize the importance of our findings and to highlight their broader implications (page 1, lines 29-36, and page 2, lines 1-50).

Comment 2

"Introduction: a good and detailed general introduction was given about the Notch signalling cascade. Since the described phenomenon is complex, I recommend to add a schematic figure about it to help your reader."

Response:

Thank you for your suggestion. We agree that a visual representation of the Notch signaling pathway can enhance understanding. However, since many review articles already provide detailed schematic figures of the Notch signaling pathway, we have chosen to refer readers to these comprehensive sources rather than including a redundant figure in our manuscript.

To improve clarity, we have enhanced our Introduction by providing more detailed descriptions of the key components and steps of the Notch signaling pathway (page 3, lines 66-87). Additionally, we have referenced review articles with illustrative figures for readers who may seek a visual representation (Page 3, lines 75-77).

Comment 3

"Materials and methods: a very detail-oriented part of the paper. No big issues were detected, however, please add the manufacturer of each chemical used (as it was nicely done in Section 4.6)."

Response:

We have updated the Materials and Methods section to include the manufacturer details for each chemical used throughout the study, ensuring consistency with Section 4.6.

Comment 4

"Results: a good description was given. Regarding figures, please use Figure 1a, 1b, 1c, etc to help your reader. Figure legends needed to be fixed, since in the current form it is hard to discriminate if it belongs to the text, or to the figure. As some genes show different expression pattern between brain parts, please do site-specific gene expression analysis."

Response:

We have revised the figure numbering to include sub-figures for clarity and modified the figure legends layout to make them more distinct and separate from the main text.

Regarding the suggestion for site-specific gene expression analysis, we have carefully considered this approach. However, due to current constraints in resources and time, we are unable to perform these additional experiments at this stage. In the results section, we have described the tissues in detail, compared the expression patterns with published literature, and cited the relevant references. Additionally, as an alternative, we have expanded the discussion section to include a thorough analysis of how regional differences in gene expression might impact our results and interpretations. Additionally, we have created Table 1, which provides a comprehensive comparison of her gene expression in the mib1ta52b mutation, highlighting regional variability and responses to Notch signaling perturbation. We hope that these alternatives will address your concerns and enhance the overall quality of our manuscript.

Comment 5

"Discussion: this is the weak point of your manuscript. Now a general overview is given about the results, but we do not gain any piece of information how these findings are integrated into the current knowledge about the topic. The importance of these findings are also missing (i.e: their role in brain development or in disease pathogenesis)."

Response:

We acknowledge this weakness and have added three paragraphs to expand the Discussion section to integrate our findings into the current body of knowledge (page 14, lines 514-524, page 16, lines 565-580, and page 17, lines 581-591). We have also discussed the implications of our results for brain development and disease pathogenesis, highlighting the importance and potential impact of our study.